# Natural Killer Cell-Mediated Immunotherapy for Leukemia

**DOI:** 10.3390/cancers14030843

**Published:** 2022-02-08

**Authors:** Michaela Allison, Joel Mathews, Taylor Gilliland, Stephen O. Mathew

**Affiliations:** 1Department of Microbiology, Immunology & Genetics, UNT Health Science Center, Fort Worth, TX 76107, USA; allison@my.unthsc.edu; 2Texas College of Osteopathic Medicine, UNT Health Science Center, Fort Worth, TX 76107, USA; joelmathews@my.unthsc.edu; 3Baylor Scott and White Sports Therapy and Research at The STAR, Frisco, TX 75034, USA; taylor.gilliland@bswhealth.org

**Keywords:** leukemia, natural killer (NK) cells, cancer, immunotherapy

## Abstract

**Simple Summary:**

Conventional therapies such as chemotherapy and radiation in leukemia increase infection susceptibility, adverse side effects and immune cell inactivation. Natural killer (NK) cells are the first line of defense against cancer and are critical in the recognition and cytolysis of rapidly dividing and abnormal cell populations. In this review, we describe NK cells and NK cell receptors, functional impairment of NK cells in leukemia, NK cell immunotherapies currently under investigation including monoclonal antibodies (mAbs), adoptive transfer, chimeric antigen receptor-NKs (CAR-NKs), bi-specific/tri-specific killer engagers (BiKEs/TriKEs) and potential targets of NK cell-mediated immunotherapy for leukemia in the future.

**Abstract:**

Leukemia is a malignancy of the bone marrow and blood resulting from the abnormal differentiation of hematopoietic stem cells (HSCs). There are four main types of leukemia including acute myeloid leukemia (AML), acute lymphoblastic leukemia (ALL), chronic myeloid leukemia (CML), and chronic lymphocytic leukemia (CLL). While chemotherapy and radiation have been conventional forms of treatment for leukemia, these therapies increase infection susceptibility, adverse side effects and immune cell inactivation. Immunotherapies are becoming promising treatment options for leukemia, with natural killer (NK) cell-mediated therapy providing a specific direction of interest. The role of NK cells is critical for cancer cell elimination as these immune cells are the first line of defense against cancer proliferation and are involved in both recognition and cytolysis of rapidly dividing and abnormal cell populations. NK cells possess various activating and inhibitory receptors, which regulate NK cell function, signaling either inhibition and continued surveillance, or activation and subsequent cytotoxic activity. In this review, we describe NK cells and NK cell receptors, functional impairment of NK cells in leukemia, NK cell immunotherapies currently under investigation, including monoclonal antibodies (mAbs), adoptive transfer, chimeric antigen receptor-NKs (CAR-NKs), bi-specific/tri-specific killer engagers (BiKEs/TriKEs) and future potential targets of NK cell-based immunotherapy for leukemia.

## 1. Introduction

Although approximately 14 billion dollars are spent annually on leukemia care alone in the US, 5-year patient mortality rates have remained consistent over the past 30 years at ~35% [1,2]. In 2022, a projected 60,650 new cases will be diagnosed, with an estimated 24,000 patient deaths [3]. These alarmingly high statistics present a pressing need to develop new therapies in order to improve treatment strategies and subsequent patient recovery [4]. Treatment for leukemia patients can vary depending on the type of leukemia, the age of the patient and stage of cancer development but will often include chemotherapy and radiation therapy [5,6]. While effective in some cases, both chemotherapy and radiation therapy are aggressive forms of treatment that can cause toxic side effects, leading to significant damage in not only the target cells, but also healthy tissues [7,8]. Recently, several forms of immunotherapy options for leukemia have been marketed including targeted antibodies, adoptive cell therapy and immunomodulators [9]. In contrast to the non-specific cell destruction induced by chemotherapy and radiation, immunotherapy offers a targeted, antigen-specific treatment option that utilizes the patient’s own immune system to combat cancer proliferation [10,11]. NK cell treatments specifically are a topic of interest as these immune cells inherently possess anti-tumorigenic characteristics and appear to present both an effective and potentially less toxic option [12,13].

### 1.1. Leukemia

Leukemia is characterized as multiple malignancies that affect the blood and bone marrow and is often the result of both genetic and environmental factors [14]. During leukemia, excess proliferation of leukemic cells from hematopoietic stem cells (HSCs) occurs, resulting in a crowding out of developing immune cells and decreased production of important lymphoid or myeloid cells. Leukemic cells are also capable of eventually infiltrating from the bone marrow into the bloodstream and have the potential to infect systemically, affecting both the peripheral and central nervous systems [15].

There are four main types of leukemia including chronic myeloid leukemia (CML), acute myeloid leukemia (AML), acute lymphoblastic/lymphocytic leukemia (ALL) and chronic lymphocytic leukemia (CLL). While these four types of leukemia are the most common and will be the primary topics of discussion, it is important to note that there are several other rare forms of leukemia as well, including prolymphocytic leukemia (PLL), large granular lymphocytic leukemia (LGL) and hairy cell leukemia (HCL). The four more common types of leukemia are separated not only by acute vs. chronic presentation, but also by the specific lineage the leukemia originates from [14]. Acute leukemia is typically more severe as cancerous cells in this state proliferate rapidly, preventing blood stem cell maturation, while chronic leukemia progresses more slowly and prevents development of blood cells from HSCs. Myeloid leukemia develops from potential myeloid cells that would typically differentiate into red blood cells, platelets or white blood cells including granulocytes or monocytes such as neutrophils, basophils and eosinophils. In contrast, lymphocytic leukemia develops from potential lymphocytic cells that would, under normal circumstances, differentiate into T cells, B cells and NK cells [16].

Conventional modalities of treatment for leukemia include chemotherapy, targeted therapy, and radiation treatment with possible bone marrow transplantation (BMT) if needed. Although these treatment models have shown various success rates in the past, patients are subject to multiple adverse side effects, increased risk of infection and the possibility of leukemic relapse. Patients with a previous history of chemotherapy or radiation therapy also have an increased risk of developing secondary leukemia, which is often difficult to treat and has a poor prognosis [16]. 

While overall 5-year survival rates for leukemia remain high, treatment of relapsed leukemia remains a major obstacle as these patients often present with chemoresistance and a significantly impaired immune system [17,18]. CAR-T cell therapies have been explored for use in leukemia and have shown some success in recognition of cancer cells, but this form of treatment is often coupled with severe side effects including encephalopathy, coagulopathy, hypoxia, and neurotoxicity [19,20,21,22]. With the success of targeting the immune system for improved patient recovery, less toxic alternatives to CAR-T therapies are being actively explored. Natural killer cell dysfunction has been well characterized in leukemia and shown to play a role in both disease severity and progression [23,24]. NK-cell based immunotherapies in the form of adoptive transfer and immune checkpoint inhibitors (ICIs) have shown promise for treatment of both primary and relapsed leukemic presentation [25,26,27].

#### 1.1.1. Myeloid Leukemia

Acute myeloid leukemia (AML) often occurs due to chromosomal abnormalities or mutations in the genes NPM1, CEPBA, RUNX1 and FLT3 and is characterized by an over-accumulation of abnormal cells called myeloblasts. AML occurs more frequently in older populations with the average age of diagnosis reported as ~65 years of age and the current 5-year survival rate stands at 28% [28]. This reportedly older age of diagnosis presents complications as some physicians may have reservations using traditional, rigorous treatment methods for elderly and subsequently more at-risk patients [28]. Treatment phases for AML typically includes induction therapy to destroy the leukemia cells followed by consolidation to kill any remaining cancer cells. Anthracycline is often the standard of care for AML, which is a drug that increases the patient’s risk of congestive heart failure. Even though rigorous treatment methods are used, 40–60% of patients relapse after initial treatment, requiring subsequent follow-up therapy in the form of hematopoietic stem cell transplants (HSCTs), additional chemotherapy, or targeted therapies [29]. Several forms of immunotherapy are currently under investigation for treatment of AML, including bispecific antibodies, CAR-T cell therapy and NK cell therapy [30].

Chronic myeloid leukemia occurs as the result of a translocation occurring between chromosomes 9 and 22, causing an abnormal gene fusion of BCR-ABL1. This is a somatic rather than acquired mutation, meaning it can occur randomly and does not require genetic predisposition. This gene fusion causes uncontrolled cell division and blockage of apoptosis, resulting in excess production of abnormal cells from HSCs, resulting in CML [31]. In similarity to AML, CML primarily affects older populations, with over half of the diagnoses being above the age of 64 [31]. Additionally, it affects slightly more men than women and accounts for approximately 15% of leukemia cases nationwide [32,33]. 

In theory, the translocation mutation is an excellent target using tyrosine kinase inhibitors (TKIs); however, many patients fail to fully respond, with a decreased rate of recovery after each subsequent treatment [34]. Even though the majority of CML patients present with resistance and in some cases intolerance to TKIs such as nilotinib, dasatinib and bosutinib, this form of therapy remains mainstay for high-risk patients [35]. TKI treatment has been shown to have a direct effect on NK cell activity. Dasatinib has been shown to increase expression of both inhibitory and activating NK cell receptors, while Imatinib has been shown to play a primarily stimulatory role by upregulating expression of only activating receptors [36,37]. Research exploring alternative therapies for TKI-resistant CML patients has shown that NK cell transfusions have the capability to overcome this drug resistance in advanced stages of development [38,39]. Since TKIs have been identified as playing a role in regulating NK cell activity, future research focused on using TKIs in combination with checkpoint inhibitors that upregulate NK cells could prove beneficial for the improvement of treatment strategies.

#### 1.1.2. Lymphocytic/Lymphoblastic Leukemia

Acute lymphocytic/lymphoblastic leukemia (ALL) is the most common form of leukemia in children and is caused by chromosomal changes such as translocations, insertions or deletions that lead to excess cell division [40]. The most common mutations that have been identified in ALL patients include translocations in BCR-ABL1, ETV6-RUNX1, TCF3-PBX1, and MLL-AFF1 with additional mutations occurring in PAX5 and IKZF1 [41]. ALL can be further categorized based on both the chromosomal and genetic mutations observed as well as the lymphocyte subtype affected (B cell lymphocytic/lymphoblastic leukemia vs. T cell lymphocytic/lymphoblastic leukemia). While 5-year survival rates for childhood ALL patients is an encouraging 87%, this survival rate significantly decreases following relapse, emphasizing the need for treatment that returns the immune system to a more homeostatic state to prevent this relapse from occurring [42]. The current standard of treatment for pediatric ALL includes combination chemotherapy, and clinical trials are underway to determine effectiveness of combining chemotherapy with targeted monoclonal antibody treatment (blinatumomab) [43]. Since leukemia cells have the potential to spread to the brain and spinal cord, treatment often includes intrathecal chemotherapy. Stem cell transplants may also be performed if a pediatric patient does not respond well to conventional treatment [44].Continued investigation of targeted immunotherapy for ALL patients remains a priority to maximize patient recovery. ALL patients with systemic peripheral leukemia have been shown to express elevated levels of IL-15, which is involved in activation of NK cells. As NK cells have been shown to be capable of direct lysis of abnormal immature WBCs in these patients, NK cell transfusions from healthy donors show promise in improving patient prognoses [45].

Chronic lymphocytic leukemia (CLL) is the most common type of adult leukemia and while it can affect both sexes, occurs more frequently in men than women. CLL is thought to occur due to damage of genes involved in blood development and results in the production of abnormal cells that would typically differentiate into the B cell subset [46]. Several somatic mutations involved in development of CLL include ATM, TP53, RB1, BIRC3 and MYD88 as well as gene rearrangements of BCR-ABL [47]. B cell deficiency characterized by this form of leukemia causes patients to have a significantly increased risk of acquiring bacterial infections. Current first lines of treatment for CLL include chemotherapy in combination with monoclonal antibodies or other forms of targeted treatment; however, these forms of therapy are not always effective [48]. Recent research suggests that use of allogeneic natural killer cells may be the future of CLL treatment [49,50].

## 2. Natural Killer (NK) Cells

NK cells are derived from HSCs via the lymphocyte cell lineage and make up ~10% of circulating lymphocytes in the human body [51]. NK cells are an important part of the innate immune system involved in recognition of virally infected or cancerous cells and are classified as CD3^−^CD56^+^ cells that can be divided into two subtypes, including CD3^−^CD56^dim^CD16^+^ NK cells and CD3^−^CD56^bright^CD16^−^ NK cells [51,52]. CD3^−^CD56^bright^CD16^−^ NK cells are typically found in secondary lymphoid tissues including the lymph nodes and are responsible for the potent production of cytokines in response to tumor cell proliferation or viral infection [52,53]. In contrast, CD3^−^CD56^dim^CD16^+^ NK cells primarily circulate in the peripheral blood and are involved in cytolytic activity when activated, releasing perforin and granzymes onto target cells [52]. Research has shown that CD56^bright^ NK cells can differentiate into CD56^dim^ NK cells upon stimulation with peripheral tissue fibroblasts [54]. Recently, studies have shown that NK cells are capable of memory, a novel finding further emphasizing the role these innate immune cells play in cancer cell surveillance [55].

### 2.1. Natural Killer Cell Mechanism of Action

Unlike T cell activity, which is antigen-specific, NK cell activity is controlled via a balance of activating and inhibitory interactions [56]. Expressed on the surface of each NK cell is an abundant assortment of activating, inhibitory and cytokine receptors that interact with their specific ligands expressed on the surfaces of other cells including immune cells, host tissues and foreign invaders [57,58]. Under normal conditions, NK cells circulate in a latent state, neither activated nor inhibited, waiting for receptor-ligand interactions to induce stimulation. When an NK cell comes in contact with a healthy host tissue or immune cell, inhibitory receptor-ligands will interact, sending signals to the NK cell that prevent cytotoxic activity [58]. When an NK cell encounters a virally infected or cancerous cell, activating receptor-ligand interactions are employed, resulting in subsequent release of lytic granules or cytokine production to induce apoptosis of the target cell. For NK cell activation or inhibition to occur, multiple receptor-ligand interactions must occur [59]. This protective mechanism is crucial to ensure that NK cells do not mass-destruct healthy tissues and that activation can only be employed when multiple interactions deem apoptosis or cytokine production necessary. Initially, it was thought that NK cell interactions relied solely on the presence or absence of MHC I, an adaptive immune receptor expressed on the surfaces of all nucleated cells [60]. While this was later disproven as red blood cells (RBCs) do not express MHC I and NK cells do not kill them, MHC I interactions remain an important mechanism by which NK cell inhibition occurs [61,62]. Although NK cells do not destroy RBCs, recognition of cells not expressing MHC I is a common trigger of apoptosis.

In the presence of tumor cells, NK cells have two main roles: degranulation resulting in cytolytic apoptosis of unrecognized cells and cytokine release to alert circulating immune cells. NK cells primarily release cytokines such as IFN-γ, TNF-α, Granulocyte-Macrophage Colony-Stimulating Factor (GM-CSF), and IL-33. These cytokines, along with IL-4, IL-7, and IL-12 stimulate recruitment and activity of hematopoietic cells for an increased immune response [63]. Cytolytic killing of tumor cells can occur either directly by activation of NK cells to induce release of cytolytic granules (perforin, granzyme, and granulysin) into affected targets or by antibody-dependent cellular cytotoxicity (ADCC) [64,65,66]. During ADCC, secreted antibodies will bind to an antigen expressed on the surface of a target cell via the Fab region. Circulating NK cells expressing the CD16 receptor recognize the Fc portion of the antibody and bind to it, inducing cross-linking of the receptors. This cross-linking activates the NK cell, signaling for the release of lytic granules and subsequent death of the target cell [67]. Some of the notable NK cell receptors that are associated with tumor surveillance and could provide therapeutic targets for leukemia treatment are natural killer group 2D (NKG2D), DNAM-1, PD-1, 2B4 (CD244, SLAMF4), CS1(CD319, SLAMF7), LLT1 (CLEC2D, OCIL), aKIRs and iKIRs, and natural cytotoxicity receptors (NCRs) such as NKp44, NKp30, and NKp46 (Figure 1).

### 2.2. Natural Killer Cell Receptors

An activating receptor widely recognized as an important modulator of cancer proliferation is NKG2D. NKG2D is expressed on all NK cells and recognizes its ligands MHC class I polypeptide related sequence A (MICA), MHC class I polypeptide-related sequence B (MICB) and UL16 binding protein 1 (ULBP1-6) [68,69]. MICA, MICB and ULBP1-6 are typically expressed in very low concentrations in healthy cells, primarily upregulated in stress-induced or cancerous tissues to stimulate NK cell activation [70]. Unsurprisingly, research has shown that NKG2D-deficient mice present with unchecked cancer proliferation due to the ability of cancerous tissue to remain incognito and avoid recognition [70]. Research has also shown that MICA and MICB possess shedding capabilities when expressed on cancer cells in an effort to avoid immune recognition. Use of a blocking antibody has been shown to reduce this downregulatory mechanism, allowing NK cell recognition. 

A second major mediator of cancer cell recognition by NK cells is the DNAM-1 receptor, an adhesion receptor involved in activation of NK cell cytotoxicity. Ligands for DNAM-1 include CD155 and CD112, and DNAM-1-CD155 or DNAM-1-CD112 interactions stimulate apoptosis of the target cell via cytolysis [71]. The T cell immunoreceptor with immunoglobulin and ITIM domain (TIGIT), an inhibitory receptor expressed by NK cells, is also capable of interacting with CD155 and CD112, preventing lytic degranulation [72]. Blockage of TIGIT-CD112 and TIGIT-CD155 interactions has been shown to improve immune function of the recipient and the ability of both NK cells and CD8^+^ T cells to recognize and clear cancerous cells [72]. While not extensively studied in hematological cancers, PD-L1 is a ligand of interest shown to be upregulated on cancer cells of AML patients with FLT3 and NPM1 mutations [73]. PD-L1 interacts via its receptor PD-1, expressed on the surfaces of both T and NK cells, suppressing immune responses. A study using anti-PD-L1 blocking antibodies in a xenograft mouse model of AML found that improved tumor lysis occurred, implying the role this receptor plays in immune suppression [74]. Studies suggest that using an anti-PD-L1 blocking antibody may be beneficial for treatment of various types of cancer when used in conjunction with NK cell-targeted immunotherapy [75,76].

Defects in any of the nine SLAM family receptors have also been associated with immune deficiencies, implicating their role in immune cell function [77]. These homophilic receptors consist of nine distinct receptors including SLAMF1-SLAMF9 [77]. SLAMF4 (2B4) and SLAMF7 (CS1) are NK cell targets of interest for the development of NK cell immunotherapies [77]. Natural cytotoxicity receptors (NCRs) are also important regulators of cancer proliferation in humans and were identified as some of the first activating NK cell receptors cloned [78]. This family of receptors includes NKp30, NKp44 and NKp46, and while these receptors are primarily involved with activating properties, NKp44 interacts with one of its ligands, PCNA, in an inhibitory fashion [79,80]. Killer cell immunoglobulin-like receptors (KIR) are receptors expressed by NK cells that recognize HLA class I (MHC class I) ligands [81]. KIR2DS1 and KIR2DL4 are two activating receptors of interest in the context of cancer elimination, while KIR2DL1, KIR2DL2/3, KIR3DL1 and KIR3DL2 are inhibitory receptors [82]. These receptor-ligand interactions occur through an immune tyrosine-based inhibitory motif (ITIM) by phosphorylating tyrosine kinases via protein tyrosine phosphatases (SHP-1 and SHP-2) [82]. Another inhibitory receptor of interest is the CD94/NKG2A receptor, which functions by recognizing levels of HLA-class I molecules present on target cells [83]. CD94/NKG2A receptors are expressed on NK cells early in the maturation process, whereas KIRs tend to be expressed later, on more mature NK cells [83]. An additional biomarker of interest is LLT1, a CLEC2D family receptor that, when upregulated on cancer cells, acts as an inhibitory biomarker of NK cell activity upon interactions with its ligand NKRP1A [84].

NK cells also express cytokine receptors including IL-21R, IL-18R, IL-12R, IL-10R, IL-4R and IL-2R [85]. These cytokine receptors will bind cytokines secreted by other immune cells, resulting in NK cell activation [85]. In addition to inducing NK cell activation, cytokine binding to the respective receptor can also stimulate prolonged functional abilities, generating long-lived NK cells. This characteristic is especially important in the context of cancer to allow adequate cytotoxicity and hinder cancer development.

### 2.3. Immune Evasion in Leukemia

The immune system is adept at recognizing abnormal or foreign cell proliferation and eliminating it before it becomes a systemic issue. Therefore, cancer cells have had to adapt multiple mechanisms by which to evade immune cell recognition. Several ways cancer cells can do this that include antigenic modulation, inducing a tumor microenvironment, suppressing immune cell function and imitation of healthy tissues [86,87]. Due to the characteristically low mutational burden observed in leukemia patients, relatively low quantities of neoantigens are presented, resulting in minimal T cell activation [88]. In the context of NK cells, leukemia can also avoid recognition by manipulating the expression of activating and inhibitory ligands on their surface [69]. Upregulation of inhibitory ligands increases the probability that corresponding NK cell receptors will recognize this ligand and continue with circulation rather than marking the cell for apoptosis. In addition to upregulation of inhibitory receptors, cancer cells can also shed HLA molecules to avoid detection by T cells [84,89,90,91,92]. Downregulated expression of MICA/MICB receptors is another mechanism of tumor escape from NK cells, given these receptors interact via NKG2D to activate NK cell activity [93,94]. This mechanism of tumor resistance is indicative of a poor prognosis in patients with leukemia specifically. Another method of tumor escape from NK cell recognition involves NKp44 interactions via its ligands with proliferating cell nuclear antigen (PCNA) and HLA I, which have shown upregulation on cancerous cells [95]. NKp44 can act as an inhibitory or activating receptor depending on the ligand interactions involved, but interactions with PCNA or HLA I cause inhibition of NK cell activity [96].

In addition to upregulation of inhibitory receptor interactions, the tumor microenvironment can also actively suppress the immune response by secreting molecules such as transforming growth factor-β (TGF-β), which inhibits NK cell effector function and survival [97,98]. Secreted enzymes such as matrix metalloproteinases (MMPs), disintegrin and metalloproteinase (ADAM) also act to shed receptors from the surface of tumor cells, further impairing immune cell-mediated elimination of tumor cells [99].

### 2.4. Functional Impairment of NK Cells in Leukemia

Research has shown that the leukemic microenvironment induces a decrease not only in the number of active NK cells, but also in the cytotoxic and degranulation capabilities of these innate immune cells [100]. A study observing lymphocyte population subsets in CML patients vs. healthy volunteers showed that CD8^+^, CD4^+^, CD3^+^ T cell and B cell numbers remained consistent, while NK cell counts dropped significantly [100]. Patients in this study treated with imatinib, a TKI, did not see a significant improvement in either NK cell activation or degranulation capabilities, indicating a continuation of a suppressed microenvironment [100]. A study looking at NK cell function in CLL established that NK cells had defective degranulation capabilities and were maintained in a primarily hyporesponsive state, while another study using AML patients showed not only reduced NK cell degranulation capabilities, but also upregulated inhibitory receptor expression and reduced TNF-α production by NK cells [101,102]. 

A study looking at NK cell activity in B- and T-ALL patients showed that numbers were significantly decreased in both the bone marrow and peripheral blood [103]. B-ALL NK cells were shown to have significantly decreased cytotoxic capabilities compared to NK cells from healthy donors when killing assays were employed with ALL sensitive leukemia cells, indicating an immunosuppressed phenotype. Further analysis showed that NK cells from B- and T-ALL patients were predominantly from the CD56^bright^ subset, which is considered an immature precursor to the more cytotoxic CD56^dim^ NK subset. This study implies that NK cell maturation and differentiation into more cytotoxic counterparts is impaired during leukemia. Suppressed NK cell activity has also been implicated in B- and T-lymphoid malignancies that present with *MYC* oncogenic abnormalities [104]. 

Mouse experiments comparing NK cell populations in healthy vs. *MYC*-activated leukemic and *MYC*-suppressed leukemic groups discovered that the group with *MYC*-activated oncogenes had drastically reduced NK cell numbers. When *MYC* was suppressed, NK cell populations were at nearly normal levels, indicating that NK cell suppression is *MYC*-dependent. Further downstream signaling pathways implicated to play a role in suppression include STAT1/2 and type 1 IFN, which are repressed by *MYC* overexpression. A study investigating NK cell dysfunction in AML used a RAG GC KO mouse model to show that when the mice were injected with leukemic blasts in combination with NK cells, there were significantly lower numbers and impaired capabilities of these NK cells 21 days after the transfusion [105]. Wild-type mice NK cells presented with higher percent perforin, granzyme B and IFN-γ expression. Flow cytometry staining of Ki62 (a marker of proliferation) on NK cells showed that NK cells in the leukemic environment had impaired proliferative capabilities. 

Similar to previous studies, leukemia-treated mice presented with impaired NK cell maturation. microRNA miR-29b was noted as highly upregulated in NK cells from leukemia-treated mice and as this miRNA has been shown to regulate T cell activity via EOMEs and T-bet, knockdowns were performed on miR-29b and found to restore NK cell activity to heightened proliferative levels. While the mechanism behind regulation of NK cell activity in leukemia is not fully understood, it is likely that the leukemic microenvironment directly (secretion of IL-10 or TGF-β) or indirectly (overexpression of suppressive-associated genes) suppresses activity via multiple different mechanisms. Collectively, these findings suggest that NK cell dysfunction is a major proponent of the leukemia microenvironment and studies focused on targeting specific receptors involved in NK-cell-suppressive effects could maximize immune function and patient survival.

## 3. Natural Killer Cell Immunotherapy for Treatment of Leukemia

NK cell-mediated immunotherapy attempts to heighten NK cell activation via blockage of inhibitory interactions, expansion of NK cell populations and enhancement of overall function. Current types of NK cell-based immunotherapies under investigation for treatment of leukemia include adoptive transfer, monoclonal antibodies (mAbs), chimeric antigen receptor-NK Cells (CAR-NKs), and bi-specific/tri-specific killer engagers (BiKEs/TriKEs) as shown in Figure 2. Research utilizing NK-based immunotherapeutic models of therapy aim to increase cases of complete remission in patients while decreasing adverse side effects often associated with the treatment.

### 3.1. Monoclonal Antibodies

Monoclonal antibodies (mAbs) are currently one of the most common forms of NK cell-mediated immunotherapy and can be utilized for both blocking certain biomarkers and boosting NK cell function by increasing ADCC. Clinical trials are currently underway to test the efficacy of combining mAb treatment with recombinant human (rh) IL-15 in order to optimize proliferation of NK cells [106]. In addition to IL-15, a pilot trial utilizing a monoclonal antibody conjugated to IL-2 showed promise for treatment of advanced melanomas [107]. Other efforts have sought to increase binding affinity of mAbs to NK cells to increase cytotoxic activity and even override inhibitory signals being transduced from iKIRs [108].

Current studies utilizing mAbs against leukemia have targeted iKIRs, aKIRs and NCRs. Early research on an anti-KIR mAb called IPH 2101 showed both restoration and enhancement of NK cell alloreactivity in adult AML patients and significantly improved patient prognosis [109]. Lirilumab, an IgG4 anti-KIR2DL-1, 2, 3 mAb, has also shown promising results in pediatric BCP-ALL patients, and clinical trials testing efficacy for treatment of both CLL and AML showed improvements in patient recovery [109,110].

Azacytidine, a chemotherapy with anti-leukemic properties, was combined with Lirilumab for clinical trials with the intent of treating AML patients. Although both drugs have shown some success separately, combining the two did not appear to significantly improve patient prognosis and trials are still ongoing to determine efficacy [111].

Since NK cells have shown high efficacy at inducing ADCC via mAb treatment, a phase I trial combined transfusions of expanded autologous NK cells with rituximab (anti-CD20 mAb) following chemotherapy and found that patients receiving this form of treatment displayed increased NK-mediated B cell lymphoma lysis [112]. Out of the nine patients in this trial, seven had a complete response; however, due to the low sample size, additional trials must be done to confirm efficacy [112]. Additional mAb trials remain underway to determine the effectiveness of targeting biomarkers involved in inhibition of both T and NK cells to optimize activity of both lymphocyte subsets against leukemia [113].

### 3.2. Adoptive Transfer of NK Cells

Adoptive transfer of NK cells has been shown to be efficacious in clinical trials for the treatment of hematological cancers as well as some solid tumors. During adoptive transfer, NK cells are isolated from the peripheral blood, bone marrow, or umbilical cord from either a healthy donor (allogeneic) or the patient (autologous) [114]. These NK cells can be purified, then genetically modified and/or expanded with cytokines such as IL-2, IL-15, and IL-21, and then infused into the patient for increased NK cell density and function [115]. Autologous HSCT is simpler to perform, but allogeneic transfer specifically shows enhanced alloreactivity assuming mismatched KIR and HLA ligands [116]. Incompatible ligands can result in graft-versus-leukemia (GVL) disease, causing increased NK cell activity because minimal inhibition will be able to occur and allowing the infused NK cells to effectively kill host leukemic cells [117,118,119].

While T cell infusions have the potential to cause GVHD (graft versus host disease) leading to significant off-target tissue damage, NK cells can suppress this interaction by secretion of IL-10 and directly lysing host-antigen-presenting cells while simultaneously targeting leukemia [119]. This important protective feature of donor NK cells emphasizes the value of using them following allogeneic hematopoietic cell transplantations. Research investigating the protective nature of NK cells has shown that this immune cell subset is the first to recover following chemotherapy or hematopoietic transplant and has been implicated in regulation of T cell activity [120]. NK cells have also been shown to limit MHC-antigen-driven proliferation of T cells and use perforin, FasL and the potentially activating receptor NKG2D to limit the expansion and activity of conventional T cell subsets [119,121,122,123].

Research has shown that using allogeneic HSCT for NK cells increases cytotoxic effects against both AML and pediatric B-cell precursor ALL (BCP-ALL) [124,125,126]. An early clinical study for allogeneic adoptive transfer using haploidentical donors treated poor-prognosis AML patients with immunosuppressive chemotherapeutics including cyclophosphamide and methylprednisolone followed by NK infusion [127]. Over 25% of AML patients in this study achieved complete remission following adoptive transfer, with mismatched donors obtaining higher complete remission rates compared to matched, a finding indicative of NK cell alloreactive capabilities [127]. A clinical study beginning in 2009 examined the effect of high dose (10–20x higher than previously used dosages) HLA-haploidentical HCT on chemo-resistant leukemia patients and found that there was a significant reduction in AML and ALL progression following treatment [128]. While only one-third of the patients obtained complete remission following therapy, this is still significant since these patients had acquired chemo-resistance and the cancer was no longer treatable with conventional therapies [128].

While allogenic transplants have shown effectiveness in treatment of AML especially, relapse due to this form of treatment is common, emphasizing the importance of post-transplant pharmacological agents to minimize mortality rates [129]. More recent clinical studies looking at post-transplant treatment tested a variety of targeted agents and found that isocitrate dehydrogenase (for IDH mutated AML), Ivosidenib (for IDH1 mutated AML), Enasidenib (for IDH2 mutated AML) and Venetoclax (selective BCL2 inhibitor) all improved clinical outcomes of AML patients meeting the drug-specific criteria [130,131,132].

A clinical trial testing treatment of intermediate AML in pediatric patients via adoptive transfer of NK cells showed minimal improvement in patient cases, indicating that either the intermediate stage of AML development or pediatric criteria may not be receptive to this form of treatment [133]. Adoptive transfer research remains important for effective treatment as it is a promising mechanism by which to improve patient immune responses without significantly damaging off-target tissues [134]. More recently, research focused on CAR-NK cell transfusions have revolutionized the future of treatment for not only leukemia and lymphomas, but also non-hematological cancers including glioblastoma and breast cancer [135,136,137,138].

### 3.3. CAR-NK Cells

CAR-T cell therapies have shown some clinical success in refractory and relapsed B-ALL cancers. However, this form of treatment is often accompanied by neurotoxicity, cytokine release syndrome (CRS) and the potential to develop GvHD [139]. Anti-CD-19 CAR-T therapy for relapsed B-ALL patients was shown to cause both GVHD and CRS in over half the patients in the study, with two patients dying from infection and another from cardiac arrest [140]. While the reported leukemia-free survival rate at 180 days was 54%, indicating treatment efficacy, 40% of patients in the study had died either due to side effects or unsuccessful leukemia neutralization. Another study using anti-CD-19 CAR-T therapy for relapsed childhood ALL presented with similar results: toxicity and severe side effects with half of the patients relapsing following treatment [141]. Recently, research investigating the long-term success of CAR-T therapies suggests that this form of treatment may not be as successful as previously thought [142]. CAR-NK cell therapy is a more recently explored option for leukemia, and while still in its experimental phases, it has shown killing of cancer with minimal risk of toxicity or GvHD [138,143,144]. In addition to enhanced efficacy and reduced negative side effects, CAR-NK therapy offers the potential for an off-the-shelf form of treatment that does not have to be individualized (in the form of HLA typing), decreasing cost and increasing the rate at which patients have access to treatment. In vivo mouse studies of the long-term effects of CAR-NKs have suggested that this form of treatment persists for as long as a year, minimizing the risk of relapse [145].

NK cells derived from multiple different sources have been engineered for comparison of efficacy including cell lines (NK92s), the peripheral blood (PB), umbilical cord derived and induced pluripotent stem cell (iPSC) derived NK cells [146,147]. While immortalized NK cell lines such as NK92 provide a convenient option, these cells must be irradiated prior to use to prevent lethal effects, resulting in less persistent NK cells that are active in a host for a short amount of time [147]. NK92 cells are also CD16- and therefore do not induce ADCC unless engineered to do so. Although these cells present some challenges with long-term stability and lysis by host PBMCs, they are nonetheless an attractive option for engineering due to their relatively versatile characteristics and robust anti-tumor activity [148,149]. PB NK cells are found in relatively high quantity and are easy to isolate, but these cells have been identified as more difficult to engineer due to low transfection efficiency [150].

In addition, cryopreservation for long-term storage has been shown to decrease cytotoxic capabilities [151]. In contrast, umbilical cord derived NK cells can undergo cryopreservation with minimal alterations and have been shown to display high proliferative capabilities, working effectively for in vivo studies [147].

Umbilical derived NK cells also express higher levels of CXCR4, a marker associated with bone marrow migration, implicating the ability to infiltrate bone-marrow derived tumors [152]. An in vitro comparison of PB vs. umbilical-derived NKs showed similar cytotoxic capabilities against CD19^+^ CLL and ALL, but in vivo research suggests PB NKs possess limited proliferation [147,153]. A clinical trial using umbilical-derived, HLA-mismatched anti-CD19^+^ CAR-NKs for treatment of relapsed and refractory lymphoma and CLL resulted in complete remission in 73% of patients [154]. All patients in this trial displayed rapid responses to the CAR-NK therapy without significant negative side effects. CAR-NK proliferative capabilities were also monitored and discovered to continue expansion properties for up to 12 months following transfusion [154]. iPSC NK cells are currently being explored and provide both a convenient safe and, standardized population for genetic engineering. This option provides an off-the-shelf form of therapy that is quick to obtain and shows high cytotoxic capabilities against tumor cells, when used with or without CAR expression [155,156].

CAR-NK constructs have been compared to mAb therapy for treatment of leukemia and data suggest that CAR-NK cells induce higher levels of cytotoxicity and are more effective overall at tumor elimination [157]. Anti-CD20^+^ mAb treatment has been shown to improve treatment of B-cell leukemias with upregulated CD20^+^ expression by enhancing NK cell activity and overcoming tumor resistance [158]. A research study of CD20^+^ CLL xenograft models compared the efficacy of using anti-CD20^+^ mAbs vs. treatment with anti-CD20^+^ CAR-NKs and discovered that CAR-NKs had significantly higher cytotoxicity against the tumors [157].

In addition to ligand-specific CAR-NK research, genome editing of NK cells to upregulate activity is also being explored. Upregulated expression of inhibitory receptors has been well-implicated in serving an inhibitory role in NK cell activation, providing a target to determine efficacy of knocking down genes coding for receptors involved in suppression. The use of CRISPR/Cas9 has revolutionized our ability to make specific and permanent gene modifications and research focused on knocking out (KO) suppression-associated markers including ADAM17 (involved in cleavage of CD16) and PD-1 have shown success in upregulating NK cell cytotoxic abilities [159]. ADAM17 KO NK cells showed potent degranulation capabilities when ADCC assays were employed with CD20 mAb-treated Raji cells. Subsequently, PD-1 KO NK cells significantly increased survival when transfused into a mouse xenograft model of ovarian cancer. This finding suggests not only the potential of gene-edited NK cells for treatment, but also that PD-1 is a major regulator of NK cell activity in prostate cancer. Issues with gene editing of NK cells include resistance to delivery methods such as plasmid-based editing, but studies investigating nucleofection of DNA into NK cells have shown high success rates [160]. In addition, NK cells that underwent nucleofection could be cryopreserved with minimal impact on efficacy. Editing NK cells and having prepared, off-the-shelf edited options for patients depending on presentation could provide an additional form of treatment. Future studies focused on long-term effects of this form of therapy are crucial to determine potential off-target effects. While still a relatively new form of therapy, the future of CAR-NK and gene edited cells for treatment of hematological cancers is bright, with multiple clinical trials (see Table 1) being actively explored to enhance NK cell activity against leukemia.

### 3.4. Bi- or Tri-Specific Killer Engagers (BiKEs/TriKEs)

BiKEs and TriKEs are monoclonal antibodies made bi- and tri- specific for tumor antigens, respectively. These antibodies are engineered by fusing together single chain variable fragments (scFv) specific for the tumor antigens of interest. This form of treatment allows for targeting multiple antigens using a single form of therapy, and once bound to the target cell, circulating NK cells are able to recognize the BiKE or TriKE and act via antibody-dependent cell mediated cytotoxicity to induce apoptosis of their target [161,162]. Current scFv targets include CD16, CD19, CD33, PD1, CTLA4, NKG2A and NKG2D [106,121,162]. The addition of cytokine moieties such as IL-12 or IL-15 have also been used to expand NK cells prior to treatment in order to increase both density and immune function against tumor cells. In 2018, a study utilizing a CS1 × NKG2D BiKE made of an anti-CS1 scFv and anti-NKG2D scFv showed increased cytokine production and NK cell lytic activity when used to treat a mouse xenograft model of multiple myeloma [163]. While BiKEs have shown some success in inducing ADCC on target cells, research shows that, comparatively, TriKEs show superiority in regards to inducing cytotoxicity, degranulation and cytokine production. A study comparing a CD16 × CD33 BiKE to a CD16 × IL-15 × CD33 TriKE in an in vivo xenograft model of AML showed that TriKE treatment paired with an allogeneic stem cell transplantation significantly restored NK cell function and activity against AML cells [164]. While the BiKE construct showed some success as well, NK cells were not as readily activated and had poorer survival comparatively, a logical finding considering IL-15 is a strong mediator of NK cell expansion [164]. An in vitro research study combining a CD16 × CD33 BiKE with ADAM17 inhibition found that combining these two forms of treatments significantly improved NK cell activity against tumor cells [165]. Current, only one TriKE clinical trial is underway, specific for CD33+ upregulated tumors (CD16 × IL-15 × CD33) including high-risk myelodysplastic syndromes, AML, systemic mastocytosis and mast cell leukemia [166].

While Table 1 showing current NK-cell immunotherapy clinical trials for leukemia is by no means comprehensive, it provides a look into the variety of options currently being explored including CAR-NKs, TriKEs, expanded NK cell infusions with mAbs, etc.

## 4. Potential Future Targets for NK Cell-Mediated Treatment of Leukemia

Leukemia cells have demonstrated the ability to manipulate expression of NK cell receptors in order to evade recognition. Targeting biomarkers involved in these suppressive effects can be beneficial in improving NK cell responses in leukemia patients. NK cell receptors 2B4, CS1, and LLT1 have been identified as playing integral roles in regulation of NK cell-mediated cytolytic activity during cancer and may provide promising targets for treatment of leukemia.

### 4.1. 2B4 (SLAMF4, CD244)

2B4 (SLAMF4, CD244) has been characterized as a receptor involved in activation of NK cell activity and is considered a biomarker of interest in AML patients [155,167]. Two different human isoforms of 2B4 have been identified including h2B4-A and h2B4-B [168]. Both isoforms have identical cytoplasmic domains with four immunoreceptor tyrosine-based switch motifs but differ in their extracellular domain due to the differential splicing of hnRNA that result in the addition of five amino acids in h2B4-B. Both the isoforms interact via their ligand CD48, which is often upregulated on the surfaces of multiple different types of cancer [84]. Upon interaction with CD48, h2B4-A increases cytolytic activity and intracellular calcium levels in NK cells, whereas h2B4-B does not activate NK cells [168]. The activation capabilities of 2B4 rely on the protein expression of SLAM-associated proteins (SAP) [84,169]. In the presence of SAP, 2B4-CD48 interactions result in NK cell activation and subsequent release of cytotoxicity on the target cell [169]. This interaction in the absence of SAP, however, can suppress NK cell activity [168,169]. Consistent 2B4-CD48 interactions can eventually result in the downregulation of 2B4 expression, a mechanism to prevent excess NK cell activity [170]. 2B4 expression can also be impacted by the presence of immunomodulatory peptides such as alloferon, which is commonly used for increasing IFN-γ and TNF-α production and NK cell activity [171]. An in vitro study looking at the effects of 2B4-specific CAR-NK cells against K562 cells (AML) showed potent co-stimulatory benefits when engineered with an intact zeta domain [172].

### 4.2. CS1 (SLAMF7, CD319)

CS1 (SLAMF7, CD319) is a homophilic receptor with two isotypes in humans including CS1-S and CS1-L. The CS1-L isotype is capable of activating NK cell cytotoxicity and has been identified as a receptor of interest for treatment of cancers that upregulate CS1 expression. Activation of CS1 results in increased production of mRNA transcripts of Fms-related tyrosine kinase 3 ligand (Ftl3l), lymphocyte-function associated antigens 1 and 3 (LFA-1, LFA-3), TNF-α, and IL-14 [173]. These cytokines and messengers are involved in the activation of inflammatory and signaling pathways to increase immune response against tumor proliferation. Elotuzumab (Empliciti) is an anti-CS1 mAb that has shown clinical efficacy and is an FDA-approved immunotherapy for relapsed/refractory multiple myeloma (MM), a type of hematopoietic cancer caused by invasion of the bone marrow by malignant plasma cells overexpressing CS1 [173,174]. Elotuzumab increases ADCC via Fc-binding to CD16 and CS1 receptors on NK cells for increased cytotoxic response to tumor cells [175,176,177,178]. CAR-NK92 cell lines containing CS1-specific scFv models increased production of IFN-γ in MM treatment models [84,179]. BiKE constructs have also been made with anti-CS1 and anti-NKG2D scFv agents in vitro (Figure 2). These models have shown similar effects to anti-CS1 CAR-NKCs, with dose-dependent increases in NK cell cytotoxicity and cytokine production. While limited research has been conducted on CS1-targeted therapy for leukemia specifically, elotuzumab could present as an effective treatment for large granular lymphocytic leukemia (LGL), as the increased presence of CS1 in LGL tumor cells decreases pro-inflammatory cytokine release [180].

### 4.3. LLT1

LLT1 is a receptor expressed on several different cancer types including triple negative breast cancer (TNBC), glioblastoma and prostate cancer [181,182]. LLT1 interacts via its ligand NKRP1A in an inhibitory manner, preventing NK cell cytotoxic capabilities [183]. Increased expression of LLT1 implies that cancer cells actively upregulate LLT1 expression, but the mechanism by which this occurs is unknown. Since LLT1 is expressed on different cells and tissue types, it presents the possibility of cancer cells escaping the immunosurveillance of NK cells. Blocking the LLT1 receptor with an anti-LLT1 mAb has been shown to increase NK cell cytotoxicity against both triple negative breast cancer and prostate cancer [182,183]. Our studies on LLT1 expression in NK cells from pediatric ALL subjects showed significant upregulation compared to NK cells taken from healthy subjects, indicating that LLT1 may play a role in immunosuppressive effects during ALL (unpublished data). Since LLT1 was shown to be significantly upregulated in pediatric ALL patients, future research investigating how blockage of this receptor via an anti-LLT1 mAb impacts NK cell responses to leukemia cells may provide insight into whether or not this receptor provides a relevant target for treatment of leukemia.

## 5. Conclusions

The future of treatment for leukemia is bright, with the utilization of NK cells against cancer cells drastically shifting the previous modalities of effective leukemia treatment. In addition to providing an alternative therapy for patients who may not respond to conventional treatment, NK cell immunotherapy focuses on harnessing a patient’s own immune system to fight cancer proliferation, minimizing off-target effects. Ongoing clinical trials are underway to determine the effectiveness of various NK cell-mediated therapies. Future research focusing on receptor-ligand interactions involved in NK cell immunosuppression including CS1-CS1, 2B4-CD48 and LLT1-NKRP1A may prove beneficial for better understanding and treatment of leukemia. While NK cell immunotherapies have shown success independently in the form of adoptive transfer or mAb treatment, future research on combination therapies such as targeted treatment combined with TKIs could optimize treatment for relapsed and refractory forms of leukemia. Continued research investigating the multimodal mechanisms by which leukemia evades detection will lead to a better understanding of the associated microenvironment and the possibility of generating more effective treatment plans.

## Figures and Tables

**Figure 1 cancers-14-00843-f001:**
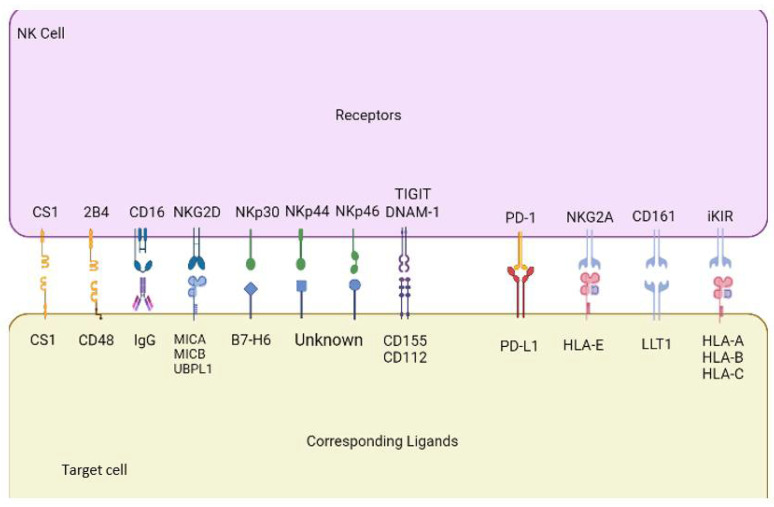
Several key NK cell receptors and corresponding ligands.

**Figure 2 cancers-14-00843-f002:**
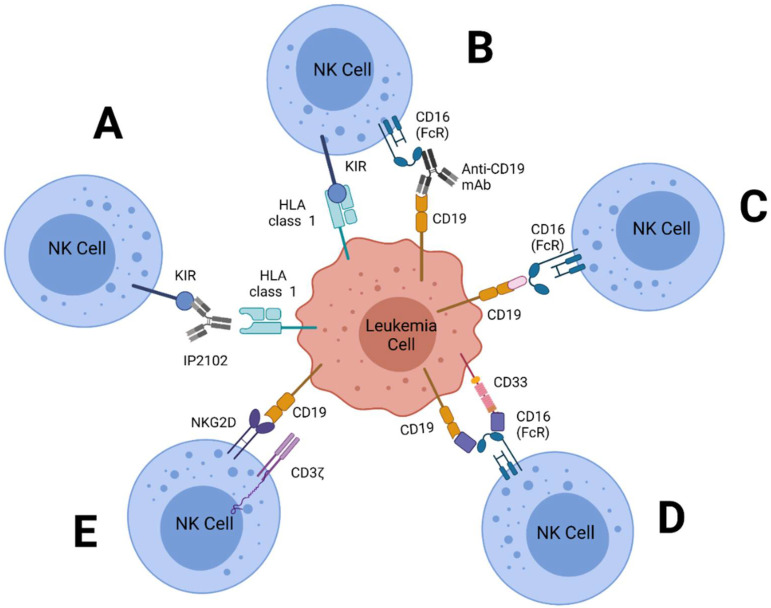
Mechanisms for increasing an NK cell response against tumor cells. (**A**) Blockage of KIR-HLA interactions by a monoclonal antibody (IPH 2102/lirilumab). (**B**) Inhibitory signals from KIR-HLA interactions are nullified by binding and activation of CD16 to monoclonal antibodies bound to CD19 antigens. (**C**) Activation of NK cells by a CD16xCD19 BiKE. (**D**) Activation of NK cells by a CD16 × CD19 × CD33 TriKE. (**E**) Utilization of CD19-recognizing CAR-NK cells with CD3ζ/NKG2D transmembrane domains.

**Table 1 cancers-14-00843-t001:** Current NK-cell immunotherapy clinical trials for Leukemia.

Treatment	Leukemia	NK Source	Phase	Status	Identifier
NKX019	B-cell ALL, CLL	Allogeneic	1	Recruiting	NCT05020678
Anti-CD33 CAR- NK + Fludarabine and Cytoxan	Relapsed/refractory AML	N/A	1	Recruiting	NCT05008575
Anti-CD19 CAR-NK + Fludarabine and Cyclophosphamide	B lymphoid malignancies	Cord blood derived	1	Recruiting	NCT04796675
CAR-NK-CD19 + Fludarabine and Cyclophosphamide	ALL, CLL, B-cell lymphoma	AT19	1	Recruiting	NCT04796688
iC9/CAR.19/IL15-transduced NK cells + Fludarabine, Cyclophosphamide or Mesna	B lymphoid malignancies, ALL, CLL	Cord blood derived	1	Active, not recruiting	NCT03056339
CAR.70/IL15-transduced NK cells + Fludarabine and Cyclophosphamide	AML	Cord blood derived	1	Not yet recruiting	NCT05092451
CD16/IL-15/CD33 GTB-3550 Tri-Specific Killer Engager (TriKE)	AML	N/A	1 & 2	Active, not recruiting	NCT03214666
NK cell infusion + Fludarabine, Cyclophosphamide and Mogamulizumab	T-cell leukemia	Third party	1	Recruiting	NCT04848064
HSCT + NK infusion + Elotuzumab and Lenalidomide or Melphalan	Plasma cell leukemia	Cord blood derived	2	Active, not recruiting	NCT01729091
FT538 NK cells + Fludarabine and Cyclophosphamide	AML	iPSC	1	Recruiting	NCT04614636
NK92 + cord blood transplant + Chemo + Rituximab	AML, ALL, CML	NK92	2	Recruiting	NCT02727803
NK cells + ALT803	AML, ALL, CML, CLL	Non-HLA matched donor	1	Active, not recruiting	NCT02890758
FT516 NK cells + Rituximab/Obinutuzumab, IL-2, Fludarabine and Cyclophosphamide	AML	iPSC	1	Recruiting	NCT04023071
oNKord + Fludarabine and Cyclophosphamide	AML	Cord blood derived	1&2	Recruiting	NCT04632316

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
