# Peer review of "Natural Killer Cell-Mediated Immunotherapy for Leukemia"

_cancers, 2022, doi:10.3390/cancers14030843_

Round 1
Reviewer 1 Report
Remarks to authors:
In this manuscript, Allison et al. describe the current research and clinical advances in harnessing NK cells for the treatment of all subgroups of leukemia.
While the Review details the diseases and lays down the different strategies of harnessing NK cells for treatment to date, it reads like an encyclopedia and does not provide any novel perspectives and fails to detail the authors’ own inferences based on the current research. The authors also do not explain the shortcomings of the current NK treatment modalities and potentially how they can be overcome. The authors have missed key recent studies which describe how NK surveillance is suppressed in leukemia. Finally, they should briefly explain where they envision the NK therapy field is headed. Just stating ongoing work without providing any insights into what niches remain to be explored in the NK therapy field make the review mundane. Major revision is therefore required.
Major points:
(1) Section 1.1: Authors should mention treatment resistance as one of the drawbacks of the current treatment modalities for leukemia. Also, because leukemia is the most common childhood cancer, and pediatric leukemia has nearly 90% 5-year survival, the authors should also briefly mention about age when they talk about current leukemia treatments and introduce NK cells. While talking about current leukemia treatments, the authors should discuss how NK cells compare to CAR-T cells.
(2) Section 1.1.2: TKI resistance of CML is an example of resistance to targeted therapies. It is not clear from the review how NK cells will overcome this? I urge the authors to provide their thoughts on this.
(3) Section 1.1.2: “Unlike CML where specific gene modifications have been identified, ALL patients present with all different types of mutations, making gene therapy a nearly impossible outlet, unless individualized.” The authors should qualify this statement with references. Please note this statement is not entirely correct because ALLs also have defined chromosomal translocations, eg. Ph+ ALL similar to CML. Please comment/ clarify further what is meant by “gene therapy” is not possible.
(4) Section 2.4: Authors have missed citing important work showing suppression of NK cell maturation in the leukemia microenvironment:
PMID: 34077953- Duault et al., Blood, 2021
PMID: 33435153- Pazina et al., Cancers, 2021
PMID: 27775550- Mundy-Bosse et al., JCI, 2016
PMID: 32503978- Swaminathan et al., Nature Communications, 2020
(5) Section 3:
- There is no discussion regarding the comparison of CAR-T cells and CAR-NK cells. CAR-T cells are extensively used for leukemia treatment and why CAR-NK would be better is not discussed.
- When the authors talk of NK cells suppressing GVHD, they have failed to cite important papers, such as PMID: 20233969- Olson et al., Blood, 2010.
- CAR-NK92 is discussed, but the advantages and drawbacks of NK-92 in comparison to healthy NK cells are not elucidated.
- The authors should discuss recent advances in CAR-NK constructs and where the NK therapy field is headed. For example, there is no discussion or mention of recent advances in NK cell genome editing.
Minor points:
- Abstract section:
In the abstract section, authors state that “Leukemia is a malignancy of the bone marrow and bloodstream”. Please replace as bone marrow and “blood”.
- Some references are incorrect. e.g. In line no. 176 “Ref number 45” does not mention cytokines secreted by NK cells.
- In section 3, sources of NK cells were repeated several times. It is recommended to make a table showing sources of NK cells (PB, cord blood, iPSE, NK cell lines), leukemia type to be treated, clinical stage of development, and preliminary findings, Ref. The same goes with monoclonal antibodies, CAR-NK cells, and B- and tri-specific NK engagers.
Author Response
Remarks to authors:
In this manuscript, Allison et al. describe the current research and clinical advances in harnessing NK cells for the treatment of all subgroups of leukemia.
While the Review details the diseases and lays down the different strategies of harnessing NK cells for treatment to date, it reads like an encyclopedia and does not provide any novel perspectives and fails to detail the authors’ own inferences based on the current research. The authors also do not explain the shortcomings of the current NK treatment modalities and potentially how they can be overcome. The authors have missed key recent studies which describe how NK surveillance is suppressed in leukemia. Finally, they should briefly explain where they envision the NK therapy field is headed. Just stating ongoing work without providing any insights into what niches remain to be explored in the NK therapy field make the review mundane. Major revision is therefore required.
Major points:
(1) Section 1.1: Authors should mention treatment resistance as one of the drawbacks of the current treatment modalities for leukemia. Also, because leukemia is the most common childhood cancer, and pediatric leukemia has nearly 90% 5-year survival, the authors should also briefly mention about age when they talk about current leukemia treatments and introduce NK cells. While talking about current leukemia treatments, the authors should discuss how NK cells compare to CAR-T cells.
Response: We highly appreciate the thorough and critical review of the manuscript. We have done extensive editing and added a lot of new material to address all of the concerns that the reviewer has raised. Specifically the following changes have been made for this query:
- Resistance was added as a major drawback to treatment in section 1.1 with cited references – lines 84-86.
- CAR-T cell effectiveness/drawbacks in treatment of leukemia are now briefly mentioned in section 1.1 and NK cell therapy introduced as a potentially effective and less toxic alternative – lines 86-90.
- NK cells are now introduced under treatment options for ALL in 1.1.2 – lines 155 – 159.
- Pediatric forms of treatment for ALL are now mentioned in 1.1.2 – lines 145-149.
(2) Section 1.1.2: TKI resistance of CML is an example of resistance to targeted therapies. It is not clear from the review how NK cells will overcome this? I urge the authors to provide their thoughts on this.
Response: Elaboration on TKI resistance and how NK cells potentially provide an effective therapy for CML has been added – lines 122 – 132.
(3) Section 1.1.2: “Unlike CML where specific gene modifications have been identified, ALL patients present with all different types of mutations, making gene therapy a nearly impossible outlet, unless individualized.” The authors should qualify this statement with references. Please note this statement is not entirely correct because ALLs also have defined chromosomal translocations, eg. Ph+ ALL similar to CML. Please comment/ clarify further what is meant by “gene therapy” is not possible.
Response: After further research, this statement has been retracted.
(4) Section 2.4: Authors have missed citing important work showing suppression of NK cell maturation in the leukemia microenvironment:
PMID: 34077953- Duault et al., Blood, 2021
PMID: 33435153- Pazina et al., Cancers, 2021
PMID: 27775550- Mundy-Bosse et al., JCI, 2016
PMID: 32503978- Swaminathan et al., Nature Communications, 2020
Response:
PMID: 34077953- Duault et al., Blood, 2021
- Article has been added to the manuscript in the NK cell dysfunction in leukemia section – lines 324 – 326.
PMID: 33435153- Pazina et al., Cancers, 2021
- While this article mentions important research on NK cell dysfunction in the hematological malignancy multiple myeloma, it is not leukemia specific. Since there is a pleura of research expounding on NK cell dysfunction in leukemia, the authors did not think this article would significantly contribute to the review so this was not included.
PMID: 27775550- Mundy-Bosse et al., JCI, 2016
- Article has been added to the manuscript and discussed in regards to NK cell dysfunction in leukemia; lines 341 – 344.
PMID: 32503978- Swaminathan et al., Nature Communications, 2020
- Article has been added to the manuscript and discussed to some degree in NK cell dysfunction section; lines 333 - 341.
(5) Section 3:
- There is no discussion regarding the comparison of CAR-T cells and CAR-NK cells. CAR-T cells are extensively used for leukemia treatment and why CAR-NK would be better is not discussed.
Response: Comparison of CAR-T and CAR-NK therapy for leukemia is now detailed in section 3.3; lines 451 – 463; 463 - 470.
2. When the authors talk of NK cells suppressing GVHD, they have failed to cite important papers, such as PMID: 20233969- Olson et al., Blood, 2010.
Response: Additional research papers including the recommended Olson et al. have been cited on NK cell role in preventing GVHD with elaboration on the mechanisms fueling this process lines; 416 - 420.
3. CAR-NK92 is discussed, but the advantages and drawbacks of NK-92 in comparison to healthy NK cells are not elucidated.
Response: More extensive elaboration with citations has been added regarding all NK sources for CAR engineering; lines 471 – 488, 501 - 504.
4. The authors should discuss recent advances in CAR-NK constructs and where the NK therapy field is headed. For example, there is no discussion or mention of recent advances in NK cell genome editing.
Response: Recent advances in NK cell gene editing is now discussed in section 3.3; lines 517 -538.
Minor points:
- Abstract section:
In the abstract section, authors state that “Leukemia is a malignancy of the bone marrow and bloodstream”. Please replace as bone marrow and “blood”.
Response: The word "bloodstream" has been replaced by "blood".
2. Some references are incorrect. e.g. In line no. 176 “Ref number 45” does not mention cytokines secreted by NK cells.
Response: References have been checked and corrected.
3. In section 3, sources of NK cells were repeated several times. It is recommended to make a table showing sources of NK cells (PB, cord blood, iPSE, NK cell lines), leukemia type to be treated, clinical stage of development, and preliminary findings, Ref. The same goes with monoclonal antibodies, CAR-NK cells, and B- and tri-specific NK engagers.
Response: Sources of NK cells for therapy have now been separated out and compared with cited studies showing effectiveness of each and also summarized in Table 1.
Reviewer 2 Report
The present review is well structured, well written and easy to read.
However, some aspect of NK cell therapies are superficially treated.
I miss a more extensive review of T-cell depleted haploidentical transplant as platform for NK cell therapy in leukemia patients and how the haploidentical transplant favor NK cell activity against leukemia.
I also miss a Table or a link where authors summarize the current clinical trial regarding NK cell-mediated immunotherapy.
Author Response
Reviewer 2 comments:
The present review is well structured, well written and easy to read.
However, some aspect of NK cell therapies are superficially treated.
1. I miss a more extensive review of T-cell depleted haploidentical transplant as platform for NK cell therapy in leukemia patients and how the haploidentical transplant favor NK cell activity against leukemia.
Response: We thank the reviewer for their thorough and critical review of the manuscript. We have done extensive modifications and added a lot of new material to the revised manuscript. Specifically, to address this query, we have elaborated more on the T-cell depleted haploidentical transplant as platform for NK cell therapy in leukemia patients and how the haploidentical transplant favor NK cell activity against leukemia in section 3.2; lines 412 - 419.
2. I also miss a Table or a link where authors summarize the current clinical trial regarding NK cell-mediated immunotherapy.
Response: As there are over 100 clinical trials exploring NK cell therapy for leukemia, this table is restricted to current CAR-NK constructs and other NK cell immunotherapy trials under investigation and has been summarized in Table 1.
Round 2
Reviewer 1 Report
The authors have addressed most of my critiques.